# Incorporating Physical Environment-Related Factors in an Assessment of Community Attachment: Understanding Urban Park Contributions

**Ying Xu** [1,2,3]**, David Matarrita-Cascante** [4] **, Jae Ho Lee** [4,]*** and A.E. Luloff** [5]

[1]  School of Geography and Tourism, Shaanxi Normal University, Chang'an, Xi'an 710119, China;
    xuying129@snnu.edu.cn
[2]  Shaanxi Tourism Information Engineering Laboratory, Chang'an, Xi'an 710119, China
[3]  Shaanxi Key Laboratory of Tourism Informatics, Chang'an, Xi'an 710119, China
[4]  Department of Recreation, Park and Tourism Sciences, Texas A&M University,
    College Station, TX 77843, USA; dmatarrita@tamu.edu
[5]  Department of Agricultural Economics, Sociology, and Education, The Pennsylvania State University,
    University Park, PA 16802, USA; ael3@psu.edu
*  Correspondence: jaeholee83@tamu.edu; Tel.: +1-979-571-9504

**Abstract:** Community sociologists have examined community attachment through an almost exclusive focus on people's social relations. Recent research efforts have noted the neglect of the physical place in traditional community sociological studies. Doing this has brought the physical environment into their discussions of community attachment. Despite this progress, we remain limited in our understanding of the physical environment's contribution to peoples' attachment to their communities and whether its effect on community attachment is applicable in the context of urban settings. In an effort to expand our knowledge of this topic, this study explored the contributions of the urban physical environment on community attachment. By selecting the Discovery Green Park as a typical form of physical environment in Houston, Texas, this study sought to investigate how people's levels of community attachment could be predicted by: (1) peoples' interactions with an urban park; (2) people's emotional connections with such a park; and (3) peoples' social interactions with others within the park. After conducting a series of block model regression analyses, we found that community attachment was not completely defined by social factors, but also depended upon peoples' emotional connections with the local physical environment and the social interactions happening in those settings.

**Keywords:** community attachment; physical-natural related factors; urban parks

## 1. Introduction

Community sociologists have long been interested in understanding social and emotional bonds to localities and the implications of such bonds for community life [1,2]. In understanding such bonds, community studies developed the notion of "community attachment" which referred to individuals' emotional investment in their community through feelings of rootedness and belonging. Previously, scholars [3,4] examined community attachment by placing a major focus on social relations (e.g., number and intensity of friendships). More recently, scholarship explored the role of the physical/natural environment in predicting community attachment [5–9]. For example, Matarrita-Cascante et al. [9] found that natural and rural-amenity motives for owning property resulted in increased community attachment for both long-term residents and seasonal residents in areas rich in natural amenities.

Sociologists have only recently begun to respond to the neglect of the physical place in traditional community sociological studies and to bring the natural environment into their discussions of community attachment [10,11]. As a result, our current state of knowledge about the processes behind the physical environment's contribution to peoples' attachment to their communities is limited. As noted by Brehm et al. [5] and Matarrita-Cascante et al. [9], the multifaceted characteristics of the physical/natural environment require further studies for us to better understand the dimensions of community attachment. Most work has focused on the natural environment in rural communities; whether urban physical environment also contributes to community attachment remains largely unstudied [12].

This study examines the contributions of urban physical environment on community attachment by analyzing the roles of peoples' physical involvement and emotional or psychological connections in urban natural settings. Here, an urban park, a typical form of physical environment in cities, was selected to address two research questions: (1) Is the influence of the physical environment on community attachment applicable in the context of urban settings? and if so, (2) In what ways does the urban physical environment contribute to individuals' community attachment?

## 2. Framework for Analysis

### 2.1. Measurement of Community Attachment

Social scientists have typically measured community attachment using one of two dominant models. The first, or linear, model held that community attachment was primarily determined by the structural characteristics of local areas. Increasing community size, population density, and heterogeneity of inhabitants associated with urbanization and industrialization acted to weaken primary bonds of kinship, which in turn decreased individuals' attachment to their communities [2,13,14]. However, numerous studies (e.g., [7,12,15]) indicated community size and population density were not significantly related to community attachment. The linear model has been criticized from theoretical perspectives and has been inconsistent in generating empirical support [12,14,16].

The second, or systemic model, viewed community as "a complex system of friendship and kinship networks and formal and informal associational ties rooted in family life and ongoing socialization process" ([2], p. 329). This model focused on the contribution of systemic factors (length of residence, social class, and age) and two groups of intervening variables (amity and associational bonds). Multiple studies tested and refined the systemic model [4,12,16,17]. Length of residence was believed to be a key variable that played a more significant role in predicting community attachment than social class and age. This reflected the fact that as individuals lived longer in a community, they had more opportunities to build relationships with other community members and become more enveloped in local social bonds. Such involvement was seen to produce greater senses of local attachment [13,18,19].

However, length of residence was not always a critical factor for predicting community attachment [10,20]. For example, Matarrita-Cascante et al. [9] found newcomers could rapidly develop a strong sentimental tie to a community based on their experience with the local natural landscapes, rather than their social bonds with other community members.

Given the importance of examining different aspects of individuals' bonds in a community, some recent research [1,6,9,10] incorporated the physical and natural landscape within the measurement of community attachment. For example, Clark and Stein [7] found high levels of community attachment were reported by natural landscape-oriented stakeholders who experienced more interactions with natural areas than socially oriented stakeholders. With a list of 14 items representing different aspects individuals perceived as being important to their community attachment, Brehm et al. [5,6] confirmed that community attachment extended beyond social aspects. Three variables—natural landscapes/views, presence of wildlife, and opportunities for outdoor recreation—were combined into a distinct environment dimension of community attachment. The authors demonstrated the importance of

including a natural environment dimension for a more complete explanation of community attachment. Building on these studies, Matarrita-Cascante et al. [9] found natural landscape-related factors were important predictors of community attachment. Regardless of length of residence, motivations of owning properties in amenity-rich areas contributed to increased community attachment. Thus, the physical/natural environment, in addition to social bonds, has been found to make a significant contribution to community attachment [6,7,9,10,21]. Building upon this research, our study expands the framework of community attachment measures by identifying physical environment-related factors. Moreover, much of the extant work in this area has focused on rural communities (e.g., [9,15]). These studies failed to examine whether physical environment related factors led to increased community attachment in an urban context. In response to the general neglect of physical/natural attributes in community sociology studies, an incorporation of the geographic place in the discussion of community attachment has occurred [11,22]. Given the significance that natural attributes may play, some scholars (e.g., [9,23]) identified a literature gap where place theory would help to better understand individuals' complex combination of feelings towards local communities.

## 2.2. Community vs. Place

Typically, place literature focused on individuals' connections with specific areas without considering the aggregate effects such connections may produce on the larger social context, such as the local communities. Conversely, community literature traditionally focused on the social bonds people developed between and among them in a local area, but ignored place-based values of where the community was located. Despite the focus and scale disparity between community and place literature [11], researchers have pointed out the potential for combining these two compatible research traditions [9,24–26]. McKnight et al. [22] and Hummon [27] emphasized a relationship between community sentiments and sense of place. Sense of place, including both cognitive and emotive components, involved people's subjective feelings toward a place and emotional reactions to that place [27]. According to McKnight, individuals inevitably imbued social meanings to the physical place where they lived and interacted with others. This helped them establish a sense of place, which in turn affected their community experience and community sentiment [22]. Hummon found varying dimensions of community sentiments provided contexts where different senses of place were formed; one was strongly related to community identity, satisfaction, and attachment [27]. This further suggested intimate relationships between people's perspectives about place and their feelings and emotions toward communities.

Given the potential for the natural environment to contribute to community attachment, Matarrita-Cascante et al. [9] suggested incorporating a sense of place construct in such studies. Two emerging aspects related to the sense of place construct may produce a better understanding of community attachment: (1) exploring the meanings people hold towards the landscape and how these meanings are tied to attachment [9,28]; and (2) determining whether attachment to a community is closely tied to one's affective feelings about a particular geographic place in the community. In pursuing its objectives, this study included the sense of place construct as a measure of emotional connections with the urban physical environment to explain if and how such connections were associated with community attachment.

## 2.3. Conceptual Model

As previous studies noting the importance of the physical/natural environment in predicting community attachment established, there is a need to explore different forms of humans' interactions and relationships with the physical environment as they lead to the creation of sentiments for a community. By and large, the predominant focus of these studies has been on amenity-rich rural areas. In contradistinction, our study asks whether the physical environment in urban areas have similar effects on community attachment. Here, using an urban park as a study site, this study specifically investigates the predictive values of different forms of human interaction and relationships associated

with the urban parks for community attachment. While a great body of literature has noted that the social bonds individuals develop with other community members are an important determinant of community attachment [4,5,12,13,16,29], this research primarily focuses on the predictive values of the physical environment.

Figure 1 shows the hypothetical model for the present study. For analyses of the predictors of community attachment, we assigned place-based sociodemographics (tied to the respondents' ZIP codes) to the first block of measures in order to verify the relative predictive power of the place-based items in urban settings. (A side note: By classifying and then comparing features between ZIP codes in the City of Houston (e.g., transportation availability, distance to the park), we were able to utilize Houston's City Limits as a large-scale community in an urban setting.) We also included individual sociodemographics (tied to respondents) in the second block of predictors as background factors intimately associated with community attachment. Following earlier research [6,9,10], the physical environment-related factors were added: we investigated the predictive values of different forms of human interaction and relationships associated with the urban physical environment on community attachment by including interactions with the environment of an urban park (the third block of predictors), emotional connections with the park (employing the measurement of sense of place; the fourth block of predictors), and social interactions that took place within the urban park (the fifth block of predictors).

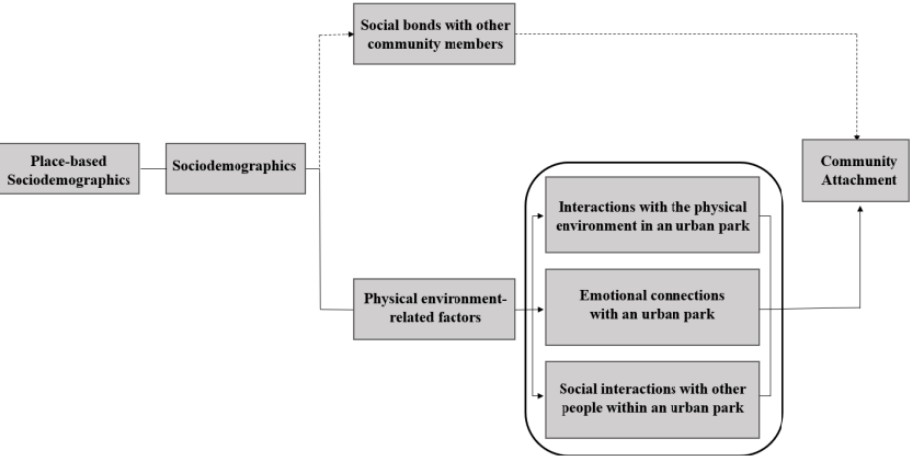

Note: The dashed line indicates the relationship between social bonds with other community members and community attachment. This relationship was not examined in this study

**Figure 1.** Conceptual model of contributions of urban park landscape-based factors to community attachment.

Three hypotheses guided this research (Figure 1):

**Hypothesis 1 (H1):** *Community attachment levels are positively associated with respondents' interactions with the environment in urban parks.*

**Hypothesis 2 (H2):** *Community attachment levels are positively associated with respondents' emotional connections with urban parks.*

**Hypothesis 3 (H3):** *Community attachment levels are positively associated with respondent's' social interactions with other people within urban parks.*

## 3. Methods

### 3.1. Sampling and Data Collection

The research study site was the Discovery Green Park in Houston, Texas. Discovery Green's 12-acres includes a variety of lawns, gardens, trails, walkways, shady areas, oak trees, and water features, while providing a large, open, *green* space in Houston's hyper-dense city center. Since it opened in April, 2008, Discovery Green has attracted more than 1.2 million visitors annually [30]. According to the Kinder Houston Area Survey, conducted annually by sociologist Dr. Stephen Kleinberg (Rice University), by the year 2013, about 26 percent of all residents in the 10 counties forming the Greater Houston Metropolitan Area had visited Discovery Green. Discovery Green benefits the local community by providing numerous opportunities for leisure pursuits and serving as a site which facilitates social interactions. Both contribute to feelings of belonging, familiarity, and attachment. Because of these reasons, Discovery Green was a suitable study site for exploring whether and how urban residents became attached to a local community based on the following: their engagement in recreational activities at Discovery Green, their emotional connections established with the park, and their social interactions with other park visitors.

The population of this study was Discovery Green park users. To attain a representative sample of these users, data was collected on weekdays and weekends during peak-and off-hours (provided by Discovery Green Conservancy) at multiple locations throughout the park in July, 2015. During each sampling period, park visitors were randomly approached and informed about this study's research objective and survey procedures. People who agreed to participate were given a printed survey questionnaire to complete.

Respondents were given an option of completing an online survey if they did not want to complete the hard copy survey at the park. Those wishing to do the online survey provided their email addresses. We then mailed a survey invitation their email accounts. The online version was created using Qualtrics and had exactly the same questions as those surveys distributed on sites. Two sequential emails were sent at intervals of 10 days [31]. For each household, only one survey was collected, completed by a family member 18 or older.

The Discovery Green Conservancy helped promote the study in several ways. First, a blog about the study was posted on the park's webpage. Second, each week during July, 2015, reference to the study was made in emails sent to park members. Third, several survey signs were made to include a short link to the survey (through the Bitly URL shortening service), and were placed in different spots in the park. Finally, a link to the online survey was included in messages on Discovery Green's Twitter and Facebook page.

As Table 1 indicates, a total of 733 people were approached on site and asked to participate in this study. Six hundred people agreed to participate and were given a printed copy. Of these, 546 surveys were returned at the park; 20 were incomplete. Among those approached at the park, 21 participants preferred to take the survey online and shared their email addresses. Messages could not be delivered to two email accounts. One person completed the survey through a link sent to his/her email account. The other 18 could not be identified by their responses (we considered these cases as non-responses). A total of 112 people (responding to the requests sent by Discovery Conservancy via social media or signs in the park) opened the online survey, and 80 of them completed the survey. The option of completing an online survey was provided primarily for those who were interested but not willing to take the time to do it while at the park. The printed copy and online version had exactly the same questions; we merged the data collected from both on-site and online and treated them equally. The overall responses to both on site and online survey were 606 (a 71.7% response rate).

**Table 1.** Survey response rate.

| | People Approached in the Park | 733 | |
|---|---|---|---|
| | Surveys distributed | 600 | |
| Printed survey | Surveys returned in the park | 546 | |
| | Non-usable surveys | 20 | |
| | Surveys completed | 526 | (71.8%) |
| Online survey | Surveys started | 112 | |
| | Survey completed | 80 | (71.4%) |
| Total | Total response | 606 | (71.7%) |

*3.2. Measurement*

3.2.1. Dependent Variable

The dependent variable for this research was respondents' perceived level of community attachment. (We introduced the place-based factors to control the impact of geographic disparities within a large-scale urban community, and we defined Houston City Limits as a large-scale community in an urban setting; in the survey, the definition of community is place-based rather than sociological, and the definition of was clearly provided to the respondents). Following Matarrita-Cascante et al. [9], we used a scale which included the following five statements: (a) "I feel this community is a real home to me"; (b) "I feel I belong to this community"; (c) "I feel I am fully appreciated as a member of this community"; (d) "Most people in this community would go out of their way to help me if I was in trouble"; and (e) "Most of the people in this community can be trusted". Respondents were asked to indicate their levels of agreement or disagreement with each of these statements using a five-point scale ranging between 1 = strongly disagree, 2 = disagree, 3 = neither disagree or agree, 4 = agree, and 5 = strongly agree.

The structure of community attachment has identified that all five items are unidimensional [9]. To confirm the factor structure, Confirmatory Factor Analysis (CFA) was performed on this construct. Each corresponding item was added until all these five items were included in the model. LaGrange multiplier (analogous to forward stepwise regression) tests were performed when each corresponding item was added [32], and one error covariance was identified. The error covariance was then removed through application of the Wald test (analogous to backward stepwise regression; [33]) so as to not produce a significant chi-square change (less than 3.84 per degree of freedom at the alpha level of 0.05; [34]). The resulting indices indicated the model fit these data very well (root mean square error of approximation, RMSEA = 0.0097, Comparative Fit Index, CFI = 0.98), although the chi-square was significant at 0.01 due to the large sample size. The community attachment construct had high internal consistency, as maximal weighted internal consistency reliability was 0.94.

3.2.2. Independent Variables

In order to measure individuals' interactions with the physical landscape, and to comport with existing literature, we included multiple independent variables:

Place-based sociodemographic variables: To take into account the statistical impact of geographic disparities within a large-scale urban community (Houston City Limits was used to define urban communities in this paper; see Figure 2), we built a data set of place-based variables (i.e., housing characteristics, public transportation, and distance to the park) by ZIP code. Despite their imperfection, ZIP code data provide plausible boundaries for the local residential environment [35]. Based on the ZIP code recorded by each respondent, we created measures of housing characteristics using the 2013-2017 American Community Survey (ACS) at the ZIP code level (ZIP code tabulation area, or ZCTA). This variable was used because it is effective for indicating an overview of neighborhood characteristics [36]. To examine data for housing characteristics by ZIP code, we first determined that the highest percentage of the total population were living in single unit residences (57.90%).

We then expressed the variable as a percentage of a total for each ZIP code. Next, to account for the potential effects of transportation accessibility to the park on community attachment, multiple shapefiles for the most common transportation modes to Discovery Green (i.e., rail stations, bus routes, and bike routes) were gathered via City of Houston GIS (COHGIS) and overlaid onto a file containing Houston's City Limits (see Figure 2). We coded these modes from 0 (no access) to 3 (access to all). Finally, we used distance to the park as a variable in order to capture potential effects of residential location [37]. For this, we used information on distance between ZIP codes and Discovery Green (77010) to aggregate the data set at the ZIP code level by distance to the park (1 = 0–4.99 miles; 2 = 5–9.99 miles; 3 = 10–14.99 miles; 4 = more than 15 miles). These four measures were used as our place-based sociodemographic variables.

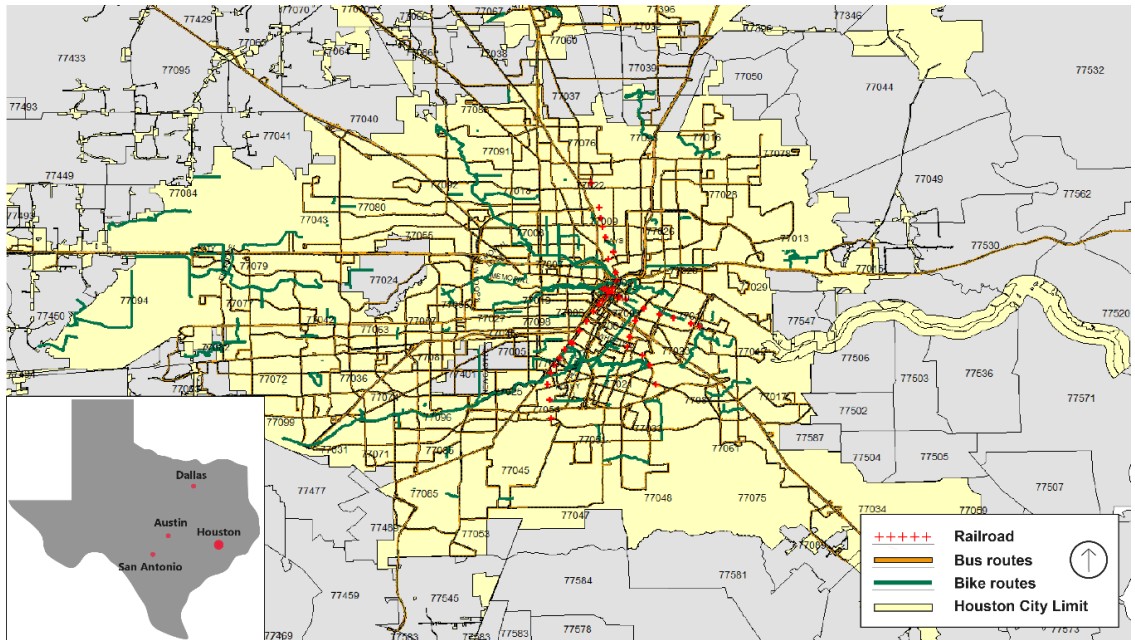

**Figure 2.** Area map of public transportation (rail stations, bus routes, and bike routes) overlaid with the Houston City Limits and ZIP codes. Transportation data was collected from City of Houston GIS (COHGIS).

Sociodemographic variables: The study gathered respondents' sociodemographic information, including age, gender, education, current employment status, and household annual income. Respondents' age was measured in years and treated as an interval variable. Gender was measured and coded as a dichotomy (0 = male, 1 = female). Education consisted of six categories: (1) less than a high school degree, (2) high school degree or GED (General Educational Development Tests), (3) some college, (4) trade/technical/vocational training or associate degree, (5) four-year college/university bachelor's degree, and (6) advanced degree (Master's, Ph.D., JD, MD). Employment status consisted of 10 categories: (1) employed for wages, (2) self-employed, (3) out of work and looking for work, (4) out of work but not currently looking for work, (5) a homemaker, (6) a student, (7) military, (8) retired, (9) unable to work, and (10) other, and coded as dichotomy (0 = unemployed (3, 4, 5, 8, 9, 10), 1 = employed (1, 2, 6, 7)). Household annual income was coded into 10 categories: (1) less than US$10,000, (2) US$10,000 to US$14,999, (3) US$15,000 to US$24,999, (4) US$25,000 to US$34,999, (5) US$35,000 to US$49,999, (6) US$50,000 to US$74,999, (7) US$75,000 to US$99,999, (8) US$100,000 to US$149,999, (9) US$150,000 to US$199,999, and (10) US$200,000 or more.

Interactions with the physical environment in an urban park: In the literature that addresses people's interactions with physical/natural areas [9,38,39], outdoor recreation engagement was an often-used indicator. Building from such studies, we designed a question that asked respondents about their type and intensity of park use. Based on the site observations and information provided by the park

authority, a sum of activities that people usually engaged in the park was created. The questionnaire included a list of 13 recreational activities: (1) attending concerts/movies/shows, (2) special events/festivals, (3) socializing with family and/or friends, (4) receptions/parties, (5) outdoor sports and/or games, (6) fitness classes (Bum-ba Toning, Yoga, Zumba, etc.), (7) visiting gardens, (8) walking the dog, (9) kayaking/boating, (10) playing around the fountain area, (11) walking/jogging/running, (12) no specific activity, just enjoy a nice day out in the park, and (13) children's programming and/or play. Respondents were asked to indicate how often during the last 12 months they had engaged in each of these activities at the park using a five-point Likert scale that ranged from 1 = never, 2 = rarely, 3 = sometimes, 4 = often to 5 = always. Principal component analysis (PCA) using direct oblimin rotation was performed with these 13 activities to identify an underlying structure. Using a cutoff value for factor loading of 0.4 (A side note: Following normal practice [40,41], cutoff values for factor loading should be above at least 0.4 in order to be considered important.), four items (4, 5, 8, 9) were deleted. As a result, three factors were extracted from the PCA: "passive activities (3, 7, 11, 12)", "park-sponsored activities (1, 2, 6)", and "children-oriented activities (10, 13)". These factors explained a total of 68.5 percent of variance, and the Cronbach's alpha values for the three factors were 0.80, 0.76, and 0.69, respectively, indicating an acceptable internal consistency for each factor.

Emotional connections with the urban park: Our conceptual model (Figure 1) also examined people's emotional connections with the park's environment by employing measurements of sense of place that included two constructs of place meaning and place attachment. Borrowing from the literature on place meaning [23,42], we asked respondents to assess the symbolic meanings local users attributed to the park. We asked respondents to indicate their levels of agreement or disagreement with each of 10 statements (using a five-point Likert scale that ranged from 1 = strongly disagree to 5 = strongly agree) about how Discovery Green park as a place was for them. The statements included 10 potential meanings that park users might hold for the park: (1) a place to escape the pressure of urban life, (2) a place to appreciate the beauty of nature, (3) a place to participate in outdoor recreational activities, (4) a place for citizens' well-being, (5) a place to meet friends and socialize, (6) a place that develops positive feelings about the community, (7) a place representing the image of Houston, (8) a place for tourists to visit, (9) a window into the diversity of Houston traditions, and (10) a fun place for children to play. According to a cutoff value for factor loading of 0.4, one item (10) was deleted, and PCA using direct oblimin rotation revealed a single dimension which accounted for 54.2 percent of the variance among these items. Reliability analysis generated a Cronbach's alpha value of 0.89, which indicated high internal consistency.

To measure place attachment, we used two scales used by Kyle et al. [43] that offered two subdomains—place identity and place dependence with four items in each dimension [43,44]. The four statements measuring place identity included: (1) "This park means a lot to me", (2) "I am very attached to this park", (3) "I strongly identify with this park", and (4) "I have special connections to this park and the people who visit the park". The four statements measuring place dependence included: (1) "I enjoy visiting this park more than any other park", (2) "I get more satisfaction out of visiting this park than from any other park", (3) "Visiting this park is more important than visiting any other park", and (4) "I would not substitute other parks for the activities I do here".

Respondents were asked to indicate their level of agreement or disagreement with each of these eight statements (four for each scale) using a five-point Likert scale ranging from 1 = strongly disagree to 5 = strongly agree. CFA was performed to test the validity and reliability of measures of the place attachment construct. The chi-square for this model was 76.0 with 19 degrees of freedom, and the *p*-value was less than 0.01. Usually, a non-significant chi-square value indicates a well-fit model. However, the chi-square value is sensitive to sample size such that it tends to be significant with large samples even though the model is fit [34]. Thus, due to this study's large sample size ($n = 606$), the significant value of chi-square was acceptable. The root mean square error of approximation (RMSEA) was 0.0071, less than 0.1, and the value of the Comparative Fit Index (CFI) was above 0.95 (0.97), which further indicated the good fit of this structural model of place attachment [34,45].

Social interactions with other people within the park. The third independent variable hypothesized as a predictor of community attachment in this study was the level of social interactions inside the park between respondents and their family members, friends, and other local park users. Some previous studies (e.g., [45,46]) suggested that visiting an urban park without company was awkward and might dissuade users from establishing social contacts there; most people tended to visit parks in groups for social gathering and relaxing. Thus, the survey first asked respondents' average group size during park visits to investigate people's motives to visit urban parks for social interactions. Respondents were then asked to indicate how often they socially interacted in the park with each of the following groups: (1) family and friends in my household, (2) family members outside my household, and (3) friends outside my household. Responses included: 1 = never, 2 = rarely, 3 = about once a year, 4 = several times a year, 5 = about once a month, 6 = several times a month, 7 = about once a week, and 8 = several times a week. PCA using direct oblimin rotation with these 3 items revealed a single dimension ("interactions with family and friends") which accounted for 76.6 percent of the variance. Reliability analysis generated a Cronbach's alpha value of 0.85, which indicated high internal consistency.

Finally, respondents were asked to indicate how often they interacted with unknown people inside the park under each of six scenarios: (1) talking with other parents when children were playing together, (2) chatting with other dog owners when walking a dog, (3) simply saying hello in passing to others, (4) playing informal sports or games with others, (5) talking with other people while attending concerts, dance parties or other special events, and (6) talking with other people while exercising, participating in fitness classes, or other programming offered in the park. Since interactions with unknown people were less common than those with family and friends in urban parks (cf., [47]), responses to this question only included five categories: 1 = never, 2 = rarely, 3 = sometimes, 4 = often, and 5 = always. PCA using direct oblimin rotation with these items was performed, and using a cutoff value for factor loading of 0.4, two items (1, 3) were deleted. As a result, a single dimension was extracted from the PCA: "meaningful interactions with unknown people (2, 4, 5, 6)". The factor explained a total of 59.7 percent of variance, and the Cronbach's alpha value for the factor was 0.77, indicating an acceptable internal consistency.

### 3.3. Data Analysis

We first report descriptive statistics of the respondents' sociodemographic variables. Then, in order to meet the study's objectives and test its proposed hypotheses, block model regression analyses were performed.

## 4. Results

### 4.1. Profile of Respondents

The sociodemographic profile of the study participants included their gender, age, education level, employment status, and annual household income (see Table 2). Almost three of every four respondents were female (72%) and the rest were male (28%). The modal respondent age category was 30 to 39 years (35.5%), followed by 19 to 29 years (31.1%), and 40 to 49 years (18.3%), indicating the majority of respondents visiting Discovery Green were relatively young. Nearly one in four respondents (24.4 %) had completed a four-year college/university bachelor's degree, followed by those who attended college but had not earned degrees (21.3%), and those with high school degrees or GEDs (19.3%). In addition, 67.7% of respondents were wage earners (employed for wages, self-employed). Slightly over than one in five respondents (20.8%) earned $50,000 to $74,999, followed by those who earned $35,000 to $49,999 (15.1%), and those who earned $100,000 to $149,999 (12.3%). The remainder of the respondents (38.8%) earned less than $50,000 with 16.7% earning less than $25,000

**Table 2.** Sociodemographic profile of respondents.

| Variables | Frequency (*n*) | Percent (%) |
|---|---|---|
| Gender | | |
| Male | 168 | 28.1 |
| Female | 429 | 71.9 |
| Age (years) | | |
| 19–29 | 187 | 31.1 |
| 30–39 | 214 | 35.5 |
| 40–49 | 110 | 18.3 |
| 50–59 | 38 | 6.3 |
| 60–69 | 20 | 3.3 |
| >70 | 5 | 0.8 |
| Education | | |
| Less than a high school degree | 27 | 4.5 |
| High school degree or GED | 116 | 19.3 |
| Some college | 128 | 21.3 |
| Trade/technical/vocational training or associate degree | 74 | 12.3 |
| Four-year college/university bachelor's degree | 147 | 24.4 |
| Advanced degree (Master's, Ph.D., JD, MD) | 106 | 17.6 |
| Employment status | | |
| Employed for wages | 355 | 55.6 |
| Self-employed | 73 | 12.1 |
| Out of work and looking for work | 30 | 5.0 |
| Out of work but not currently looking for work | 14 | 2.3 |
| A homemaker | 70 | 11.6 |
| A student | 45 | 7.5 |
| Military | 1 | 0.2 |
| Retired | 11 | 1.8 |
| Unable to work | 3 | 0.5 |
| Other | 10 | 1.7 |
| Annual household income | | |
| less than $10,000 | 44 | 7.3 |
| $10,000 to $14,999 | 26 | 4.3 |
| $15,000 to $24,999 | 31 | 5.1 |
| $25,000 to $34,999 | 42 | 7.0 |
| $35,000 to $49,999 | 91 | 15.1 |
| $50,000 to $74,999 | 125 | 20.8 |
| $75,000 to $99,999 | 57 | 9.5 |
| $100,000 to $149,999 | 74 | 12.3 |
| $150,000 to $199,999 | 46 | 7.6 |
| $200,000 or more | 32 | 5.3 |

*4.2. Block Model Regression Analysis*

The block regression model used to predict community attachment included five sequential blocks: (1) place-based factors, (2) sociodemographics, (3) interactions with the park's environment, (4) emotional connections with the park, and (5) social interactions with others inside the park (see Table 3). After entering place-based factors and sociodemographics as control variables (Models 1 and 2), the block regression model included three sequential blocks (A side note: To rule out the existence of common method biases in a data set, Harman's single factor test was conducted, and no significant issue emerged. The results are not presented but are available upon request from the corresponding author.): interactions with the park's environment, emotional connections with the park, and social interactions with others inside the park.

**Table 3.** Hierarchy regression analysis for predictors associated with community attachment (standardized coefficient).

| | Model 1 | | Model 2 | | Model 3 | | Model 4 | | Model 5 | |
|---|---|---|---|---|---|---|---|---|---|---|
| **Place-based factors** | | | | | | | | | | |
| Housing type (×2) | −0.016 | | −0.030 | | −0.029 | | −0.022 | | −0.018 | |
| Transportation (×3) | −0.184 | | −0.192 | | −0.188 | | −0.129 | | −0.145 | |
| Distance to the park (×4) | −0.395 | *** | −0.401 | *** | −0.387 | *** | −0.291 | ** | −0.303 | ** |
| **Sociodemographics** | | | | | | | | | | |
| Age (×5) | | | 0.026 | | 0.036 | | −0.005 | | −0.013 | |
| Employment (×6) | | | −0.033 | | −0.025 | | −0.019 | | −0.016 | |
| Income (×7) | | | 0.137 | * | 0.111 | | 0.125 | * | 0.129 | * |
| **Interaction with the physical environment** | | | | | | | | | | |
| Passive activities (×8) | | | | | 0.006 | | −0.065 | | −0.062 | |
| Park-sponsored activities (×9) | | | | | 0.107 | | 0.033 | | −0.012 | |
| Children-oriented activities(×10) | | | | | 0.095 | | 0.073 | | 0.073 | |
| **Emotional connections with the urban park** | | | | | | | | | | |
| Place meaning (×11) | | | | | | | 0.157 | * | 0.150 | * |
| Place identity (×12) | | | | | | | 0.257 | ** | 0.261 | ** |
| Place dependence (×13) | | | | | | | 0.076 | | 0.077 | |
| **Social interactions with other people** | | | | | | | | | | |
| Group size (×14) | | | | | | | | | 0.009 | |
| Interactions with family and friends (×15) | | | | | | | | | −0.091 | |
| Meaningful interactions with unknown people (×16) | | | | | | | | | 0.124 | * |
| Df | 3 | | 6 | | 9 | | 12 | | 15 | |
| $R^2$ | 0.071 | | 0.091 | | 0.117 | | 0.265 | | 0.282 | |
| F | 7.341 | *** | 4.782 | *** | 4.168 | *** | 8.431 | *** | 7.272 | *** |

\* $p <0.05$, \*\* $p <0.01$, \*\*\* $p <0.001$.

The first model tested the predictive values of place-based sociodemographic variables including housing characteristics, public transportation, and distance to the park. Results from Model 1 showed that only distance to the park was significantly and negatively associated with community attachment ($\beta = -0.395$, $p < 0.001$). This meant that people who lived closer to the park were more attached to the local community. This first model explained 7.1 percent of the variance.

The second model tested the predictive values of sociodemographic variables including age, employment status, and household income. Controlling for the place-based variables, only income ($\beta = 0.137$, $p < 0.05$) was significant to community attachment. This meant that people who earned a higher income were more likely to attach to the local community than people who earned less. Distance to the park, which was significant in the previous model at the level of 0.001, remained significant ($\beta = -0.401$) at the same level. This model explained 9.1 percent of the variance and was significant with an F-ratio of 4.78 ($p < 0.001$).

In Model 3, variables of interactions with the park's environment were introduced. Controlling for the place-based and the sociodemographic variables, none of the interaction with the physical environment was significantly related to attachment. Remaining as significant in this model was distance to the park ($\beta = -0.387$, $p < 0.01$). This model explained 11.7 percent of the variance and was significant with an F-ratio of 4.17 ($p < 0.001$).

Model 4 introduced the emotional connection variables including place meaning, place identity, and place dependence. Controlling for the rest of the variables, place meaning ($\beta = 0.157$, $p < 0.05$) and place identity ($\beta = 0.257$, $p < 0.01$) were significantly and positively related to community attachment. Respondents who attributed more symbolic meanings to the park landscape and strongly identified with the park were more attached to the local community. Distance to the park, which was significant in the previous model, remained significant ($\beta = -0.291$, $p < 0.01$) and income attained significance ($\beta = 0.125$, $p < 0.05$). Adding the emotional construct substantially increased the proportion of explained variance beyond that contributed by place-based factors and sociodemographics. This model explained 26.5 percent of the variance, and was significant, F=8.43 ($p < 0.001$).

The social interaction variables were introduced in Model 5. Controlling for the rest of the variables in the model, meaningful interactions with unknown people was significantly and positively associated with community attachment ($\beta = 0.124$, $p < 0.05$). This means that people who carried out meaningful interactions with unknown people tended to have higher levels of community attachment. In this model, distance to the park ($\beta = -0.303$, $p < 0.01$) and income ($\beta = 0.129$, $p < 0.05$) remained statistically significant, as well as the two emotional connection variables that were significant in the previous model—place meaning ($\beta = 0.150$, $p < 0.05$) and place identity ($\beta = 0.261$, $p < 0.01$). Adding the emotional construct substantially increased the proportion of explained variance beyond that contributed by place-based factors, sociodemographics, and emotional construct. Model 5 explained 28.2 percent of variance and the F-ratio was 7.27 ($p < 0.001$).

## 5. Research Findings Discussion

This study explored the physical realm within community attachment research in an urban setting. Its aim was to examine the effects of physical environment-related factors on community attachment.

We hypothesized that respondents' community attachment was positively associated with: (1) their interactions with the physical environment in an urban park (H1); (2) their emotional connections with the park (H2); and (3) and their social interactions with others inside the park (H3). The results revealed that Hypothesis 1 failed to be supported, while Hypothesis 2 and 3 were partially supported. In the case of Hypothesis 1, when the effects of the rest of the variables in the model were controlled, no statistical significance in the model was found. Our study results did not support the hypothesis that respondents' interactions with the physical environment in an urban park were positively related to community attachment levels. These findings were consistent with the conclusion of Matarrita-Cascante et al. [9], who reported that residents' levels of community attachment in amenity-rich rural areas were not

significantly explained by participation in recreational activities. Thus, recreational engagement may not be associated with community attachment in either urban or rural settings.

Concerning Hypothesis 2, our analysis found two dimensions of respondents' emotional connections with an urban park were positively associated with community attachment. These consisted of the constructs measuring place meaning and place identity. Place meaning was a significant factor predicting respondents' community attachment levels. According to Stedman [25,28], place meanings are cognitions people hold towards certain places, and represent a descriptive statement of what a place means. Humans attribute symbolic meanings to a place based on their cognition, and further develop a positive emotional or affective bond to it [23,48]. In this study, Discovery Green was an important everyday place which local families and individuals often visited. While the usages of Discovery Green varied by individuals (e.g., walking, picnic, children's programming), the park was fundamental to daily life for local residents; through their experiences, these residents ascribed symbolic meanings toward the park. Combining with individuals' range of usage and experiences in the park, these formed the evaluative judgments of the place, which found purchase in their increased levels of attachment to the local community.

In addition, place identity was a strong, positive predictor of community attachment. Respondents' attachment to the local community was strongly tied to their self-identity with the park. This follows earlier theoretical arguments that one way to lead to increased community attachment is to promote place identity associated with the physical environment [49]. It has been suggested that humans naturally harbor physiological attachment to natural objects [50,51]. Here, residents' emotional attachments developed from their experiences at the park and contributed to their community attachment. This may especially be the case in a compact urban area with high density in an inner city, like the inner-Houston area studied here. In the case of Discovery Green, most residents seemed to have favorable images of the park. As an iconic feature of Houston, the park provided them with diverse cultural events and recreational programs, as well as a sense of civic pride and identity. People's appreciation of physical attributes in Discovery Green evoked feelings towards the park and enabled residents to connect urban life with their self-characteristics. Such feelings found expression in residents' emotional sentiments, and their feelings of belonging and identification with local communities.

Nevertheless, actual behaviors, like participation in recreational activities, have been found to be unrelated to community attachment. The physical environment serves as a basis of community attachment with people who may have less direct experience with the environment yet feel strong emotional connections. This is in line with the place attachment view that people can form strong bonds to places that they have never visited or directly experienced [23,52–54]. Thus, we may conclude that individuals can develop strong attachment to their community due to their emotional bonds with certain local environments regardless of their direct experience with that environment. Here, emotions played a role more important than actions in determining community attachment.

Our research findings partially supported Hypothesis 3 in that a significant relationship between meaningful interactions with unknown people in the park and community attachment was found. Consistent with the relevant literature [1,9], the physical environment was important for determining community attachment with reference to social interactions occurring in physical settings. Respondents who had interacted with strangers in Discovery Green while walking dogs with other dog owners or participating in some recreational activities provided by the park (e.g. sports, concerts, dance parties, fitness classes), reported higher levels of local sentiments. This was because park visitors who had a perception that other people had a reason, such as sharing the same interests with them, were more likely to share the space with them and to establish contact [46]. A combination of knowing others and having cursory interactions in public places has previously been found to stimulate feelings of comfort and help put people at ease [47,55]. Visitors engaging social activities in a public space had more extensive social ties with other visitors [46]. By providing gathering places in congested cities, parks facilitated social interactions and collaboration [56]. These interactions further led to increased social networks and social cohesion, giving support to the formation of community attachment.

Besides the predictive power of individuals' different types of interactions with the park environment, an objective measure of distance to the park appeared to be a robust significant predictor that was negatively related to community attachment levels. Individuals who lived closer to the park area reported higher levels of attachment to the local community. This was in line with Arnberger and Eder's [10] finding that residents' perceptions of nearby public green spaces was related to higher community attachment. Previous research highlighted that proximity to urban parks and green space was positively related to physical activity, public health, and community wellbeing [57,58]. Citizens living adjacent to the park could be both more aware of its various offerings as well as received the benefits associated with proximity to the public park, and, as a result, exhibited a greater sense of community attachment.

In addition to environment-related predictors, one sociodemographic variable—household income, was found to be positively and significantly related to community attachment. People of higher income exhibited higher levels of sentiments toward the local community, as reported previously (e.g., [8,9,16,59]). Compared to their counterparts, affluent residents usually had more interest in local affairs. Moreover, their higher income allowed them to establish and stress local social bonds, as well as allowed them to have more opportunities to pursue local community goods and services. Together these helped to improve their life satisfaction and social psychological well-being, which led to greater attachment to local communities.

## 6. Conclusions and Implications

This study sought to examine the roles of physical environment in predicting community attachment in order to improve our understanding of the physical/natural dimension of community attachment. On a more theoretical level, to refine the approaches of previous research [5], this study expanded the theoretical framework by targeting multiple environment-related factors: respondents' interactions with the environment through engagement of recreational activities, respondents' spiritual and emotional bonds to the physical environment, and social interactions that occurred within physical settings. While prior studies focused on the predictive qualities of the physical/natural environment in amenity-rich communities, this research shifted the discussion to an urban setting. Regardless of the different conditions and functions of the physical environment in rural and urban contexts, this study found common ground with earlier studies with respect to the potential contributions of the physical environment on increased community attachment. Further, it revealed that community attachment was not completely socially dominant, and also depended on individuals' emotional connections with the local physical environment and the social interactions that happened in such physical settings.

In addition, our study provided insight on the link between place and community theories to better understand relationships between people and locales. Utilizing measures of sense of place, we have empirically supported the theoretical supposition proposed by Stedman et al. [25] that cognition and evaluation of the natural attributes of locales (place meaning and place identity) serve as a basis for increased attachment at the community level. Consistent with previous research (e.g., [60]), this study found that attachment to place was often tied to respondents feelings about their community. This study's findings demonstrated that adding measures of sense of place would contribute to a better understanding of how people develop strong attachment to their community with reference to the local physical environment.

This research also has important practical implications for urban park planners and designers. For many urban residents, parks are very important in their day-to-day life. Citizens perceive urban parks in different ways based on their experiences at these places and resulting connections they develop with them. The strong levels of place meaning and place identity found in our study were indicative of the deep emotional connections residents developed with Discovery Green. It is crucial that park planners and designers understand how people use parks and become connected with them and ensure diverse viewpoints are incorporated into their proposed planning efforts. As noted by Farnum et al. [61], if people deeply connect their self-identity with places, it is important for managers

to be aware that their actions should not interfere with people's interactions with those places in ways that result in negative effects on visitors' place-based experiences.

Additionally, the recognition of place identity associated with the urban park in this study provides managers with valuable knowledge when evaluating changes and modifications in the design and management of parks. Individuals who strongly identify with a particular place are usually sensitive to changes associated with those places. This suggests that urban park planners and managers should implement design and management strategies progressively, allowing time to evaluate these changes and modifications, and revise those decisions by taking the public's responses into account [62].

It is also important to consider the impacts of urban parks on community attachment by stimulating social interactions. Although most visitors to urban parks are in the company of family and friends, interactions with other unknown people while engaging in some activities in park areas enable them to develop social networks. Parks are of significance not only in initiating social interactions, but also in sustaining such social ties [46]. This stresses the importance of designing urban parks with adequate recreational activities to increase the potential of social interactions.

Considerations of the emotional connections between local users and urban parks and social interactions that occur in such places should be integrated into urban park planning strategies and policy decisions. These considerations could be better understood by facilitating discussions on the public's needs and value orientations associated with their parks. According to Worpole and Knox [63], the success of a public space not only relies on the urban architects, planners, and designers, but also depends on the people who visit and use the space. The establishment of strong people–park bonds would contribute to the survival and sustainability of urban parks by forming cooperation and collaborations between communities and park agencies.

As with all research, there are limitations. First, the study is limited to one study site—Discovery Green Park. No comparisons can be made to determine whether the predicting variables examined in this study will present consistent or different powers in other urban parks. Parks have different characteristics in terms of size [64], number of features and amenities [65], presence of sports fields [66], trails [65,67,68], and accessibility [69]. All these attributes are associated with use of parks and engagement of physical activity at parks [70] for different groups of people. This suggests that parks may elicit varying levels of preference and connections. It would be fruitful for future research to include additional study sites to explore whether these physical environment-based factors display consistent predictive values on community attachment across parks.

Second, this study is limited to a park environment only. Broadly thinking, the urban physical environment consists of various types such as green spaces, green trails, and gardens. People enjoy different types of environments for different purposes, which in turn shapes their evaluations and attitudes associated with them [71]. Given the multifaceted characteristics of the urban physical environment, future research should incorporate multiple types of urban environments, examine their effects on community attachment, and discern whether such effects vary across these various environments.

Third, more attention needs to be given to measurement-related issues as perhaps indicated by the insignificant relationships found between community attachment and interactions with the park's physical environment. Our study measured the interactions through respondents' participation in several recreational activities at the park. These measures, however, are somewhat abstract, and were not significant in predicting levels of community attachment. Here, we did not investigate respondents' levels of involvement in urban park activities, their motivations for park visitation, or satisfaction with their experience associated with the park environment and qualities of park services. However, we believe incorporating such measures would present a more complete conception of how urban physical environment-based interactions contribute to increased community attachment. Thus, in future studies, additional measures are needed to better explain the outcome variable.

In summary, we believe this study makes significant contributions to the community literature through its exploration of the role of the physical environment in forming community attachment.

Our results suggested that emotional connections with the local physical environment and the social interactions that occurred there were important determinants of how residents became attached to their communities. Usually, attachment reveals its importance when it brings people together to work collectively to address issues of collective concern [38]. From our perspective, a better understanding of the sources of community attachment is central to promoting participation—a critical component of successful community development [9]. Community leaders who recognize the importance of the physical environment in residents' attachment to their communities are better equipped to implement effective development strategies, those guided by community interests to promote and sustain local environmental qualities.

**Author Contributions:** Conceptualization, Y.X.; Data curation, J.H.L.; Formal analysis, J.H.L.; Investigation, Y.X. and J.H.L.; Methodology, Y.X. and D.M.-C.; Supervision, D.M.-C. and A.E.L.; Visualization, J.H.L.; Writing—original draft, Y.X.; Writing—review and editing, D.M.-C. and A.E.L.

**Funding:** The open access publishing fees for this article have been supported by The National Natural Science Foundation Project, China, the 'Study on Place Integration of Tourism Immigrants in Ethnic and Religious Communities in the Typical Cities of Western China' (Grant No. 41671144), and the Fundamental Research Funds for the Central Universities (Grant No. GK201903078).

**Conflicts of Interest:** The authors declare no conflict of interest.

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
