# Peer review of "Incorporating Physical Environment-Related Factors in an Assessment of Community Attachment: Understanding Urban Park Contributions"

_sustainability, doi:10.3390/su11205603_

Round 1

Reviewer 1 Report

Abstract

Change “…their communities and whether the effect of such environment on community attachment are…” to “… their communities and whether the effect of such environment on community attachment is…”

Introduction

Changes are underlined: “…two research questions: (1) is the influence of the physical environment on community attachment applicable in the context of urban settings? and (2) if so, in what ways does the urban physical environment contribute to individuals’ community attachment?

Framework for Analysis

Community vs. Place:  Following this sentence, “McKnight et al. and Hummon emphasized a relationship between community sentiments and sense of place”, please provide readers with more information on what defines a “sense of place”.  For example, how did McKnight et al. and Hummon measure sense of place?  Since a sense of place construct was included in the analysis, this is a good place to give readers a description of that construct.

Conceptual Model:  Please delete “there is” in this sentence: While there is a great body of literature has noted that social bonds… 

The word “block” alone might confuse readers – maybe say the “first block of predictors”, the “second block of predictors”, etc.

As written, the sentence “Following earlier research [6,9,10], the physical environment-related factors were added” makes it a little unclear as to whether the physical-environment related factors were part of the second or third block.  I suggest changing this to the following:

“Following earlier research [6,9,10] the physical environment-related factors were added: we investigated the predictive values of different forms of human interaction and relationships associated with the urban physical environment on community attachment by including interactions with the environment of an urban park (the third block), emotional connections with the park (borrowing the measurement of sense of place; the fourth block), and social interactions that took place within the urban park (the fifth block).”

Regarding the note at the bottom of figure 1, please change to read as: “Note: The dashed line indicates the relationship between social bonds with other community members and community attachment.  This relationship was not examined in this study.”

Methods

Independent variable:  General question: A “cutoff” value of .4 – what is this cutoff value for… factor loading?  Please clarify.

Results

Why did the explained variance in dependent variable decrease from 26.4% to 24.4% from model 4 to model 5?

Discussion

The Discussion should contain the following elements and typically (but not always) in the following order:

Restate the purpose and summarize the results.

Indicate whether hypotheses were supported (if there are hypotheses for the study)

Compare, contrast, and explain why the results agree or do not agree with results of previous literature (i.e., literature cited in the Introduction).

Outline theoretical and practical implications.

Identify strengths and limitations.

Provide recommendations for future research.

Summarize and state conclusions.

The Discussion overall looks very good, but I did not see content related to number 5 and, with the exception of the last sentence in paragraph two of the Discussion, I did not see content related to number 6.  Please add some content for number 5 and a bit more for number 6.

Author Response

We would like to sincerely thank this reviewer for her/his comments that facilitated a significant improvement to the quality of the manuscript.

Point 1: Change “…their communities and whether the effect of such environment on community attachment are…” to “… their communities and whether the effect of such environment on community attachment is…”

Response 1: The suggested changes have been incorporated in the manuscript.

Point 2: Changes are underlined: “…two research questions: (1) is the influence of the physical environment on community attachment applicable in the context of urban settings? and (2) if so, in what ways does the urban physical environment contribute to individuals’ community attachment?

Response 2: The suggested changes have been incorporated in the manuscript.

Point 3: Community vs. Place: Following this sentence, “McKnight et al. and Hummon emphasized a relationship between community sentiments and sense of place”, please provide readers with more information on what defines a “sense of place”.  For example, how did McKnight et al. and Hummon measure sense of place?  Since a sense of place construct was included in the analysis, this is a good place to give readers a description of that construct.

Response 3: We have addressed this point by adding additional information about sense of place in the manuscript.

Point 4: Please delete “there is” in this sentence: While there is a great body of literature has noted that social bonds… 

Response 4: It was a mistake we created, and we deleted “there is” in the manuscript. Thank you for catching it.

Point 5: The word “block” alone might confuse readers – maybe say the “first block of predictors”, the “second block of predictors”, etc.

Response 5: The suggested change has been incorporated in the manuscript.

Point 6: As written, the sentence “Following earlier research [6, 9, 10], the physical environment-related factors were added” makes it a little unclear as to whether the physical-environment related factors were part of the second or third block.  I suggest changing this to the following:

“Following earlier research [6, 9, 10] the physical environment-related factors were added: we investigated the predictive values of different forms of human interaction and relationships associated with the urban physical environment on community attachment by including interactions with the environment of an urban park (the third block), emotional connections with the park (employing the measurement of sense of place; the fourth block), and social interactions that took place within the urban park (the fifth block).”

Response 6: The suggested changes have been incorporated in the manuscript.

Point 7: Regarding the note at the bottom of figure 1, please change to read as: “Note: The dashed line indicates the relationship between social bonds with other community members and community attachment. This relationship was not examined in this study.”

Response 7: The suggested change has been incorporated in the manuscript.

Point 8: Independent variable:  General question: A “cutoff” value of .4 – what is this cutoff value for… factor loading? Please clarify.

Response 8: We apologize for the lack of a clear explanation. We clarified as follows: “Using a cutoff value for factor loading of .4.”

Also, we added the text below in a footnote.

“Following normal practice [73,74], cutoff values for factor loading should be above at least .4 in order to be considered important.”

Point 9: Why did the explained variance in dependent variable decrease from 26.4% to 24.4% from model 4 to model 5?

Response 9: We acknowledged the mistake we created. As the adjusted R square increases from .234 to .244 in the table (see original text in the Model 4 and 5, not the one overlapped with the values of R square), the explained variance (R square) reported in the text should also be increased. We changed 24.4% to 28.2% in the text.

In addition, to avoid confusion between the text and the table, we reported R square in the table instead of adjusted R square.

Point 10: The Discussion should contain the following elements and typically (but not always) in the following order: 

Restate the purpose and summarize the results.

Indicate whether hypotheses were supported (if there are hypotheses for the study)

Compare, contrast, and explain why the results agree or do not agree with results of previous literature (i.e., literature cited in the Introduction).

Outline theoretical and practical implications.

Identify strengths and limitations.

Provide recommendations for future research. 

Summarize and state conclusions.

The Discussion overall looks very good, but I did not see content related to number 5 and, with the exception of the last sentence in paragraph two of the Discussion, I did not see content related to number 6.  Please add some content for number 5 and a bit more for number 6.

Response 10: The suggested changes have been incorporated in the Discussion and Conclusion sections respectively.

Reviewer 2 Report

This study examines how people’s community attachment could be explained by interactions with park, emotional connections with park, and social interactions with others within the park, using a sample of users of Discovery Green Park. 

This study exerted significant efforts to collect data and writing is sound, but the results are not convincing. 

First of all, several model results can be biased by "common method bias". Most significant results of this study are from variables measured from a single cross-sectional survey, while most ZIP-coded block-level analysis doesn't show meaningful results. For example, interactions with the environment in urban parks, emotional connections with urban parks, social interactions with other people within urban parks are all measured by survey questions, and Community attachment levels are also measured by the same survey with same respondents. Content of specific items, scale type, response format, and the general context of the study can create a common latent factor, but common latent factor was not addressed. 

Second, this is a self-selection bias problem because this study used self-selected individuals who are park users as well as who are willing to participate in the survey. Bias arises in this situation in which individuals select themselves into a group, causing a biased sample with nonprobability sampling. In other words, non-users are excluded in the sample, therefore, data is censored. The author shouldn't generalize the results/finding to the general public because of this. The data can't show the actual picture because the data excluded non-users. What if non-users who don't have an emotional connection with park have a lot higher levels of community attachment? 

Third, the result of Place-based factors is not explained theoretically. Results from Model 1 shows that distance to the park was significantly and negatively associated with community attachment. This may be an interesting finding, but I can't find a hypothesis about this in any of H1-H3 and Figure 1. And there can be many socio-economic and/or demographic factors in the distance to park, but it is unclear whether these are controlled or adjusted in the model specification. 

Figure 2 area map does not contribute to any of the research questions or purpose of this study and I don't see why this is presented. 

Author Response

We would like to sincerely thank this reviewer for her/his comments that facilitated a significant improvement to the quality of the manuscript.

Point 1: First of all, several model results can be biased by "common method bias". Most significant results of this study are from variables measured from a single cross-sectional survey, while most ZIP-coded block-level analysis doesn't show meaningful results. For example, interactions with the environment in urban parks, emotional connections with urban parks, social interactions with other people within urban parks are all measured by survey questions, and Community attachment levels are also measured by the same survey with same respondents. Content of specific items, scale type, response format, and the general context of the study can create a common latent factor, but common latent factor was not addressed. 

Response 1: We understand your concerns regarding “common method bias,” which may create high correlation among the independent variables and dependent variable. To test if common method biases exist in the data, we used Harmon’s single-factor test. The most common post hoc test allows us to check whether a single factor is accountable for variance in the data (please see more articles using this method; Chang et al., 2010; Podsakoff et al., 2003; Tehseen et al., 2017).

Following the protocols of this method, we entered all items of independent variables (interactions with the environment in urban parks, emotional connections with urban parks, and social interactions with other people within urban parks) and dependent variable (community attachment) into factor analysis. The generated PCA output (see Table 1) revealed 9 factors accounting for 69.5% of the total variance. The first unrotated factor captured only 27.4% of the variance in data. Thus, the underlying assumptions of the Common Method Bias were not met, i.e., no single factor emerged and the first factor did not capture most of the variance. Therefore, these results suggested that Common Method Bias is not a pervasive issue in this study.

Table 1.Harman’s single-Factor Test
Extraction Method: Principal Component Analysis

Component

Initial Eigenvalue

Total

% of Variance

Cumulative %

1

10.42

27.42

27.4

2

3.99

10.51

37.9

3

2.84

7.48

45.4

4

2.21

5.80

51.2

5

1.87

4.93

56.1

6

1.54

4.05

60.2

7

1.36

3.57

63.8

8

1.15

3.03

66.8

9

1.03

2.72

69.5

We added the text below in a footnote in the results section.

“To rule out the existence of common method biases in a data set, Harman’s single factor test was conducted, and no significant issue emerged. The results of this test are not presented but are available upon request from the corresponding author.”

Chang, S. J., Van Witteloostuijn, A., & Eden, L. (2010). From the editors: Common method variance in international business research.

Podsakoff, P. M., MacKenzie, S. B., Lee, J. Y., &Podsakoff, N. P. (2003). Common method biases in behavioral research: A critical review of the literature and recommended remedies. Journal of applied psychology, 88(5), 879.

Tehseen, S., Ramayah, T., &Sajilan, S. (2017). Testing and controlling for common method variance: A review of available methods. Journal of Management Sciences, 4(2), 142-168.

Point 2: Second, this is a self-selection bias problem because this study used self-selected individuals who are park users as well as who are willing to participate in the survey. Bias arises in this situation in which individuals select themselves into a group, causing a biased sample with nonprobability sampling. In other words, non-users are excluded in the sample, therefore, data is censored. The author shouldn't generalize the results/finding to the general public because of this. The data can't show the actual picture because the data excluded non-users. What if non-users who don't have an emotional connection with the park have a lot higher levels of community attachment? 

Response 2: We appreciate this comment but kindly disagree. The purpose of this paper is not to measure community attachment for those who live in the City of Houston. Rather, the purpose is to examine the roles of physical environment-related factors (Discovery Green in this article) on community attachment. In other words, this paper seeks to reveal and better articulate the contributions of physical place to community attachment research. Non-users may have different levels of attachment to their communities through social interaction or participating in a meeting for example, but the kinds of attachment this paper focuses on are tied to physical places. For this reason, we selected respondents who visited/used the Discovery Green rather than the general population in Houston including non-park users.

Point 3: Third, the result of Place-based factors is not explained theoretically. Results from Model 1 shows that distance to the park was significantly and negatively associated with community attachment. This may be an interesting finding, but I can't find a hypothesis about this in any of H1-H3 and Figure 1. And there can be many socio-economic and/or demographic factors in the distance to park, but it is unclear whether these are controlled or adjusted in the model specification. 

Response 3: We understand this comment, but, again, disagree. The inclusion of place-based factors (e.g., transportation accessibility, residential location) in the model was to control their potential effects on community attachment. Since the main idea of this paper is to investigate the roles of physical environment-related factors (i.e., interactions with the environment in urban parks, emotional connections with urban parks, and social interactions with other people within urban parks) on community attachment, we did not include place-based factors in the hypothesis.

And, with respect to the second part of Point 3, controls for socio-economic and/or demographic factors were included in the second model.

Point 4: Figure 2 area map does not contribute to any of the research questions or purpose of this study and I don't see why this is presented. 

Response 4: The purpose of the inclusion of Figure 2 was to show how we constructed the control variables, which are place-based factors (i.e., housing characteristics, public transportation, and distance to the park). We think that one area map showing public transportation overlaid with the Houston City Limits would be informative to readers to understand geographic disparities within a large-scale urban community and the importance of controlling for the variables.

Reviewer 3 Report

The paper is well-written and presents comprehensive data and analysis.  I have a few thoughts for the authors to consider:

Epistemology and methods: in community sociology, community attachment refers to people's reliance on the qualities of social interactions and relations, while in environmental psychology, place attachment or belonging is a much broader concept encompassing the social, physical and practical aspects of a place. So there is nothing for community attachment to focus on the social dimensions because it is precisely how the concept is defined. So if you want to incorporate the physical dimension into a study of community attachment, your legitimate research question is necessarily like this: how the physical dimension of the place contributes to people's perception of the qualities of SOCIAL interactions and relations in the community. Otherwise, community attachment just becomes conflated with place attachment. This research question needs to be spelled out more clearly in the paper;

2. Relatedly, when you undertook the survey how did you ensure that the respondents understood how the word "community" referred to a sociological concept rather than a place referent? Bear in mind that in Chinese language, "community" or shequ is a notoriously vague term that can mean both a sociological "community" or a physical "neighbourhood" in the English language. 

3. Measurement: the readers cannot see measurement of the qualities of physical environment or facilities in the park, such as whether the facilities are adequate, whether they are of good standards, etc. Instead, the survey variables include only those related to people's interactions and emotional attachment. The authors need to justify this omission. 

Otherwise I am fine with this manuscript. 

Author Response

We would like to sincerely thank this reviewer for her/his comments that facilitated a significant improvement to the quality of the manuscript.

Point 1: Epistemology and methods: in community sociology, community attachment refers to people's reliance on the qualities of social interactions and relations, while in environmental psychology, place attachment or belonging is a much broader concept encompassing the social, physical and practical aspects of a place. So there is nothing for community attachment to focus on the social dimensions because it is precisely how the concept is defined. So if you want to incorporate the physical dimension into a study of community attachment, your legitimate research question is necessarily like this: how the physical dimension of the place contributes to people's perception of the qualities of SOCIAL interactions and relations in the community. Otherwise, community attachment just becomes conflated with place attachment. This research question needs to be spelled out more clearly in the paper;

Response 1: First of all, we appreciate your comment. We agree that the suggested research question is relevant to the literature and may in fact attempt to address it in future research. However, our research aims were more limited in this study. Our objectives were to determine if and how social and physical elements independently play a role in community attachment. This interest was driven by earlier research suggesting that place matters in community attachment.

Point 2: Relatedly, when you undertook the survey how did you ensure that the respondents understood how the word "community" referred to a sociological concept rather than a place referent? Bear in mind that in Chinese language, "community" or shequ is a notoriously vague term that can mean both a sociological "community" or a physical "neighbourhood" in the English language. 

Response 2: At the beginning of each survey given to the respondents, we noted that “The community here refers to the Houston City Limits.” The definition of community in this study is place-based rather than sociological, and the definition was clearly presented to the respondents.

We included the text below in the manuscript

in the survey, the definition of community is place-based rather than sociological, and the definition of was clearly provided to the respondents”

Point 3: Measurement: the readers cannot see measurement of the qualities of physical environment or facilities in the park, such as whether the facilities are adequate, whether they are of good standards, etc. Instead, the survey variables include only those related to people's interactions and emotional attachment. The authors need to justify this omission. 

Response 3: The most recent literature on community attachment that has indicated the connection between physical elements and community attachment has focused on the interaction and emotional connections with the place. There is no mention on this literature on the need to quantify the quality of the place. Both the community and the sense of place literature focus on interactional measures, instead of a very structural measure of place as the reviewer suggested. Our research is driven by the relevant literature that points to interactions with place, not value judgement of its quality. Thus, the major purpose of this study is to examine the role of physical place in determining community attachment through individuals’ interactions with the place.  The reviewer’s suggestion may raise a good question to guide the future research by adding the measurements of qualities of physical environment, which however is not the objective of this study.

Round 2

Reviewer 2 Report

The authors basically disagreed all the points made in the first review, hence little-to-no changes were made in the manuscript. Therefore, my verdict is the same.